# Perfecting the Puzzle of Pathophysiology: Exploring Combination Therapy in the Treatment of Type 2 Diabetes

**Ridhi Gudoor, Austen Suits**  **and Jay H. Shubrook ***

College of Osteopathic Medicine, Touro University California, Vallejo, CA 94592, USA
* Correspondence: jshubroo@touro.edu

**Abstract:** Type 2 diabetes mellitus (T2DM) is a debilitating, lifelong condition with a rising incidence. A wide variety of antihyperglycemic agents are available on the market to treat diabetes. However, the number of patients living with diabetes under suboptimal control remains relatively high. This calls into question whether the application of the current treatment standards is effective and durable to truly manage the disease well. This paper aims to highlight the various classes of antihyperglycemic agents from a pathophysiologic perspective and explore the best possible combination that can have a durable effect on diabetes management. To determine this, an eight-piece pathophysiologic puzzle was created, each piece representing an organ system affected by the disease—liver, pancreas (alpha and beta cells), muscle, adipose tissue, gut, brain, and kidneys. Choosing a combination therapy that is both durable and can effectively address all eight pieces of the puzzle can theoretically create sustainable ameliorating effects. This combination can potentially lead to reduced microvascular and macrovascular complications, as well as work towards creating an ideal long-term, affordable diabetes care plan.

**Keywords:** diabetes; combination therapy; octet

## 1. Introduction

*Case*

A 49-year-old cis gender male colleague was recently diagnosed with type 2 diabetes (T2DM) and comes to see you to start treatment. He knew he had a family history of T2DM but was hoping to prevent this from coming on so soon. He has seen what T2DM can do in his patients and wants a progressive treatment that will prevent the typical experience of feeling always behind and the continual need to add medications. His only other known medical history is a history of obesity (BMI: 30.5) and metabolic syndrome (elevated waist circumference and dyslipidemia (HDL: 32 mg/dL; LDL: 136 mg/dL; triglycerides: 188 mg/dL)). He is on no current medications and has no known allergies. He does not follow any specific diet and tries to walk his dog at least twice a week. He has ample insurance with low co-payments and he is open to any of the current treatment regimens.

What treatment regimen would you offer this patient?

Diabetes mellitus (DM) affects nearly 11% of the United States population, with an incidence rate of 1.4 million cases per year [1]. Furthermore, it was the seventh leading cause of death in the United States in 2019 [2]. Currently, there are over 100 medications available to treat diabetes. Despite the wide variety of antihyperglycemic agents, a 2015–2018 study showed that nearly 50% of diagnosed patients with diabetes had uncontrolled hyperglycemia with an HbA1c > 7% [3]. Despite the explosion of new treatment options, glycemic control remains less than optimal.

In this article, the authors examine the efficacy of the current management of type 2 diabetes mellitus and discuss the pathophysiology behind the disease and its progression. Highlighting each pathophysiological mechanism as a piece of a puzzle, the authors explore currently available antihyperglycemic agents and the pieces of the puzzle they solve. In

search of a treatment option that solves all eight puzzle pieces, the authors review notable studies utilizing combination therapy for the treatment of T2DM and suggest an ideal combination of agents that addresses the entire puzzle of the pathophysiology to provide durable treatment focused on organ-protective endpoints.

## 2. Current Guidelines for Treating T2DM: Are They Effective?

Based on 2022 American Association of Clinical Endocrinologists (AACE) clinical practice guidelines, the current glycemic algorithm is a patient-centered, step-up approach with individualized medical nutrition therapy and individualized DM therapy based on the level of glycemia and present comorbidities [4]. While metformin remains a commonly prescribed initial therapy, it is no longer considered the only acceptable first-line agent. If the patient has established or high cardiovascular risk, heart failure, or chronic kidney disease, then GLP-1RAs or SGLT-2is are recommended [5]. If HbA1c levels remain >7.5%, then a dual-therapy combination is recommended, and if no improvement is seen in 3 months, then a triple-therapy combination is recommended [5]. Injectable therapy should be considered if the initial HbA1c levels are >9.0%, starting with a GLP-1RA followed by insulin treatment [5]. Benefits of using a step-up approach include the ability to accurately assess the effects of each new drug added to the regimen and monitor the patient's risk for adverse events [6]. However, previous trials have proven that initial combination therapy is more effective at achieving rapid glycemic control [7,8].

When considering the best treatment approach, clinicians must contemplate durability in addition to efficacy. The durability of medications may be defined as the ability to postpone or delay the progression of the disease in a safe and well-tolerated manner [9]. Clinically, this can be measured as the need for an increased medication dosage, frequency, or number of agents required to maintain HbA1c levels below the glycemic goal [9]. As the insulin secretory capacity of patients with type 2 diabetes diminishes over time, it is important to find a therapy that is both durable and protective of pancreatic beta cells [10,11].

Considering these concepts, is the application of the current diabetes management guidelines adequate to achieve effective and durable glycemic control? The trends suggest not. The authors believe that in order to create a durable effect, the answer lies in addressing the pathophysiology of the disease. It is critical to devise a treatment plan that focuses on the root of the condition rather than placing a band-aid on the problem.

## 3. Pathophysiology of T2DM: The Ominous Octet Theory

The understanding of the pathophysiologic mechanisms behind diabetes mellitus has vastly progressed over the last decade. What was originally thought to be caused solely by the dysfunction of a singular organ, the pancreas, has now evolved into a multi-organ pathway with many intricate variables [12].

One of the first multi-organ theories was introduced in 1987 by the physician Ralph DeFronzo. He utilized laboratory and clinical studies to present the concept of a triumvirate contributing to the pathophysiology of non-insulin-dependent diabetes: pancreatic beta cells, muscle, and liver [13]. According to DeFronzo, in the early stages of diabetes mellitus, muscle tissue and hepatic cells become insulin-resistant while glucose tolerance remains normal as pancreatic beta cells work to increase insulin secretion and offset any defect [13]. However, this eventually results in beta-cell exhaustion, leading to impaired insulin secretion and eventually overt hyperglycemia.

In 2009, DeFronzo further refined this theory to include five additional pathways and thereby formed the ominous octet [14]. Each of these additional pathophysiological pathways was selected after multiple studies demonstrated its contribution to glucose intolerance and thus a hyperglycemic state.

The first addition to the triumvirate was adipose tissue. The inclusion of the fat cell was based on studies demonstrating the phenomenon of lipotoxicity, which refers to increased FFA levels triggering gluconeogenesis, increasing insulin resistance, and decreasing insulin secretion [15–17]. Thus, the triumvirate became the disharmonious quartet.

Next, gastrointestinal tissues were implicated to form the quintessential quintet. This piece of the puzzle revolves around the incretin effect of GLP-1 and GIP. Studies have demonstrated that subjects with impaired glucose tolerance have decreased GLP-1 secretion and increased GIP secretion, resulting in the rise of plasma glucagon and the failure to appropriately suppress hepatic glucose production after meals [18,19].

The sixth component of DeFronzo's octet is the pancreatic alpha cell. Studies have demonstrated that patients with type 2 diabetes have elevated glucagon secretion, which further increases the basal rate of hepatic glucose production and, thus, fasting hyperglycemia [20,21].

The kidney was the next organ added to form the septicidal septet. Patients with type 2 diabetes have been shown to have significantly increased levels of SGLT2 channels, resulting in a maladaptive response of increased glucose reabsorption [22].

Finally, the octet is completed by the brain. DeFronzo and colleagues demonstrated that appetite regulation via insulin is impaired in key hypothalamic areas using fMRI. These results suggest that the brain may also be resistant to insulin and thus contribute to the pathophysiology of diabetes [23].

Collectively, the ominous octet encompasses the pancreatic beta cells, liver, muscle, adipose, gastrointestinal tissue, pancreatic alpha cells, kidneys, and brain. Each of these eight pathways described in the ominous octet revolves around two main mechanisms: declining tissue sensitivity to insulin and decreased insulin secretion [14]. From this theory, further advancements have been made to create more thorough and comprehensive frameworks regarding the underlying cause of diabetes mellitus. Examples of these theories include the Egregious Eleven and the twin-cycle hypothesis [24]. However, the authors of this paper believe that at the core of the pathophysiology that causes diabetes mellitus are the eight main pathways identified by the ominous octet.

Based on this compilation of research, it should be noted that beta-cell functionality is a critical aspect to preventing the further progression of diabetes mellitus [24]. By effectively minimizing the risk of beta-cell exhaustion, permanent damage from downstream pathways can be prevented. Choosing a medication therapy that can tackle all the relevant pathways is critical to reducing stress on beta cells.

If the ominous octet is said to highlight the main pathophysiological pathways contributing to T2DM, then an eight-piece puzzle can be created in which every piece of the puzzle incorporates an organ from the octet (Figure 1). When designing the puzzle, the triumvirate (liver, muscle, pancreatic beta cells) was placed at the top, as it is derived from the original theory, while the other pieces are the five additional factors that were incorporated later. Strategically selecting a medical therapy that can address all eight pieces of the puzzle can theoretically tackle every pathway. Before assessing viable combinations, it is important to review existing medications and assess the pieces of the puzzle that they ameliorate (Figure 2).

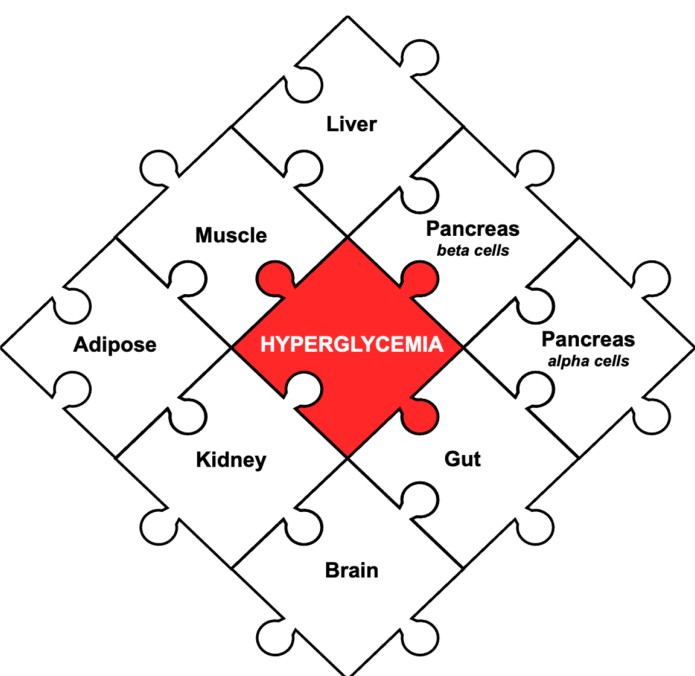

**Figure 1.** The eight pieces of the pathophysiology that contribute to hyperglycemia in T2DM. The puzzle was originally designed to incorporate all eight organs highlighted in the ominous octet. At the top is the original triumvirate: liver, pancreatic beta cells, muscle. Around it are the additional five factors: pancreatic alpha cells, adipose, kidneys, gut, and brain. In the center is the main treatment goal: hyperglycemia.

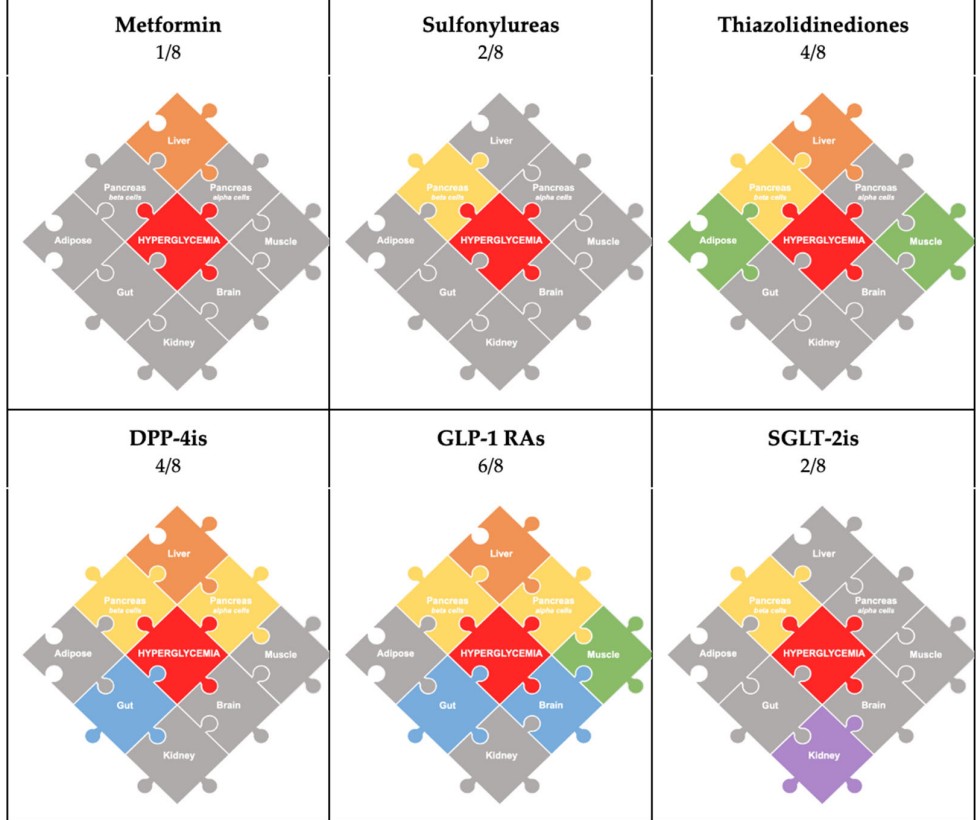

**Figure 2.** *Cont.*

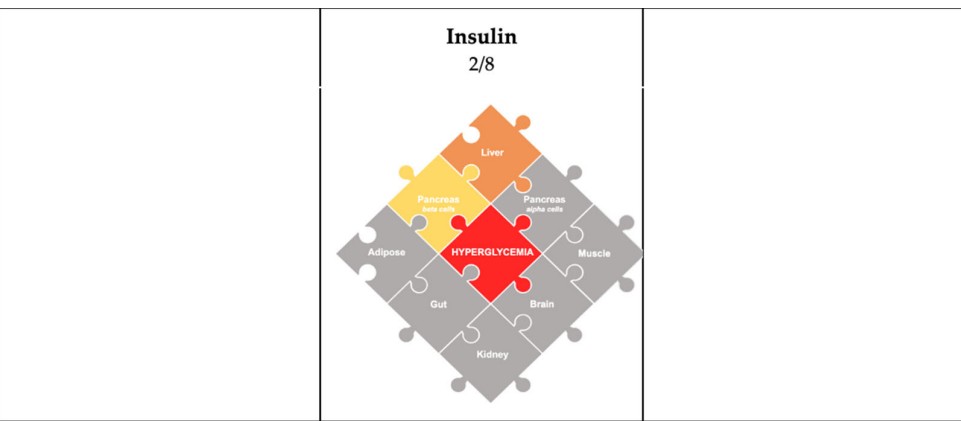

**Figure 2.** Graphic depicting the pathophysiological mechanisms addressed by common antihyperglycemic agents. Here, each individual puzzle is color-coded according to the drug class and the particular organ pathway that it addresses. The gray pieces in the puzzle represent the specific mechanisms that are not addressed by the medication.

## 4. Antihyperglycemic Agents (Figure 2: Color-Coded by Medication)

Metformin, historically the first-line medication for type 2 diabetes, works by inhibiting hepatic glucose production. Importantly, it does not have any beta-cell-protective features [25]. Therefore, it can tackle one piece of the puzzle: the liver.

Sulfonylureas, the most common second-line therapy, stimulate insulin secretion independent of glucose levels. However, limitations include a lack of protective effect on the beta-cell function and adverse effects of hypoglycemia and weight gain [26,27]. Therefore, they also tackle only one piece of the puzzle: the pancreas (beta cells).

Thiazolidinediones have a dual effect of improving insulin sensitivity in muscle, liver, and adipose tissue as well as preserving the beta-cell function [28]. In particular, pioglitazone is the preferred agent due to its cardioprotective effects and reversal of hepatic steatosis [27]. However, negative side effects associated with pioglitazone include increased weight gain, fracture risk, edema, and heart failure [29]. As a monotherapy, thiazolidinediones complete half the puzzle (4/8): the liver, muscle, adipose tissue, and pancreas (beta cells).

Incretin-based therapies act on the gastrointestinal–hepatopancreatic–brain axis and modulate gut hormones to maintain euglycemia and regulate appetite [30]. They minimize two important adverse effects seen with many other therapies: hypoglycemia and weight gain [27]. The two main classes are glucagon-like peptide 1 receptor agonists (GLP-1RAs) and dipeptidyl peptidase 4 inhibitors (DPP-4is), and a new class of dual-incretin agents has recently emerged.

DPP-4 inhibitors increase the availability of incretins (GLP-1/GIP), enhancing glucose-stimulated insulin secretion and glucagon suppression, leading to reduced hepatic glucose production [31]. Studies have shown that although they are associated with improved beta-cell function, they have no effect on insulin sensitivity and are weight-neutral [27]. Therefore, they complete 4/8 of the puzzle: the pancreas (alpha and beta cells), liver, and gut.

Pharmacologically, GLP-1RAs are more potent medications than DPP-4is. They mimic the effects of GLP-1, causing glucose-induced insulin secretion and the inhibition of glucagon secretion, and delaying gastric emptying, similar to DPP-4 inhibitors [32]. However, they have been shown to indirectly enhance both muscle and hepatic insulin sensitivity via the induction of weight loss [33,34] and exert protective effects on pancreatic beta cells [35]. Other beneficial actions attributed to GLP-1RAs include neuroprotective effects, the modulation of lipid metabolism, and protective cardiovascular effects [36,37]. As a monotherapy, they complete 6/8 pieces of the puzzle: liver, muscle, pancreas (alpha and beta cells), gut, and brain.

In 2022, the FDA approved the use of tirzepatide, a new medication that works as a dual GIP/GLP-1 receptor agonist (also known as "twincretin"), which causes a synergistic effect by enhancing the insulin response and producing a glucagonostatic effect during hyperglycemia [38]. Twincretins have similar effects to GLP-1RAs and, therefore, they also complete 6/8 pieces of the puzzle: the liver, muscle, pancreas (alpha and beta cells), gut, and brain. GIP has important functions in both regulating glucagon levels and maximizing post-prandial glucose-dependent insulin secretion.

Sodium-glucose cotransporter 2 inhibitors (SGLT-2is) have durable glucose-lowering effects via the increased urinary excretion of glucose and are associated with a weight loss of 2–3 kgs, reduced blood pressure, and positive cardiovascular and renal outcomes [29,39,40]. Studies have shown that SGLT-2is decrease glucotoxicity and result in significantly improved beta-cell function; however, there is no significant change in insulin sensitivity associated with the agent [27]. Therefore, they tackle 2/8 pieces of the puzzle: the kidneys and pancreas (beta cells). The most noteworthy aspect of SGLT-2is is that they are the only medication class that targets the renal-pathway puzzle piece in the octet.

Insulin therapy can be initiated at any time and has demonstrated the ability to achieve glycemic control [41]. A systematic review and meta-analysis of interventional studies determined that short-term intensive insulin therapy can improve the beta-cell function and decrease insulin resistance [42]. The adverse effects associated with insulin therapy include hypoglycemia and weight gain [41]. It can complete 2/8 of the puzzle as a monotherapy: the pancreas (beta cells) and liver.

## 5. Monotherapy vs. Dual Therapy

Individually, each medication class has been clinically shown to achieve glycemic control with up to a 1% reduction in the HbA1c. However, they lack durability when used as a monotherapy [25,29]. The durability of therapy can be defined by reduced disease progression [9]. Pathophysiologically, this can be measured via a worsening beta-cell function due to decreased insulin sensitivity or increased insulin resistance [9]. When using a combination of medications that do not treat the entire octet puzzle, this clinically presents as an increased drug dosage, drug frequency, or number of antihyperglycemic agents needed to maintain an HbA1c < 7.0% [9].

The current treatment of DM involves an individualized plan based on presenting comorbidities, complications, and access to medicines [4]. Per the AACE guidelines, metformin has been cited as the preferred initial therapy; however, is this truly the best medication? The 5-year ADOPT trial showed that metformin as a monotherapy had a maximal treatment effect on HbA1c levels at 12 months of therapy, and a mean HbA1c level < 7% was maintained for 45 months [25]. However, when assessed in clinical practice, metformin monotherapy was proven to be ineffective in long-term glycemic control [43]. In an observational cohort study conducted at Kaiser Permanente Northwest in 2004, 1799 drug-naive patients initiated on metformin had HbA1c levels < 7% within 3 months, yet 42% of these patients experienced secondary failure (HbA1c levels > 7.5%) within a mean period of 27.6 months [43].

The most prescribed second-line medication after monotherapy failure with metformin are sulfonylureas [44]. They reduced HbA1c levels by 1% compared to a placebo [43]; however, the UK Prospective Diabetes Study (UKPDS) 49 showed that 3 years of monotherapy using SUs had a 50% monotherapy failure rate, which further increased to 76% in 9 years [45]. The study concluded that within 3 years after the diagnosis of diabetes, approximately 50% of patients will need either a dual or combination therapy because monotherapy is not effective at achieving the target value of an HbA1c < 7% [44].

Following the failure of metformin monotherapy, AACE guidelines suggest dual-combination therapy. When used in tandem, medications have the potential to work synergistically and complete more of the puzzle, which can result in better outcomes. A comparison study between monotherapy and dual-combination therapy determined that the dual-therapy outcomes further reduced the HbA1c levels by about 1.0% compared to

monotherapy [46]. However, no dual combination can tackle all eight pieces of the puzzle when used with metformin.

## 6. Existing Combination Studies

It has been demonstrated that combination therapy remains more effective than monotherapy due to the synergistic effects of the medications that tackle various pathological targets [47]. Over the years, there have been various studies that have explored the benefits of combination therapy and their results. From the list, there are a few studies to take note of (Figure 3).

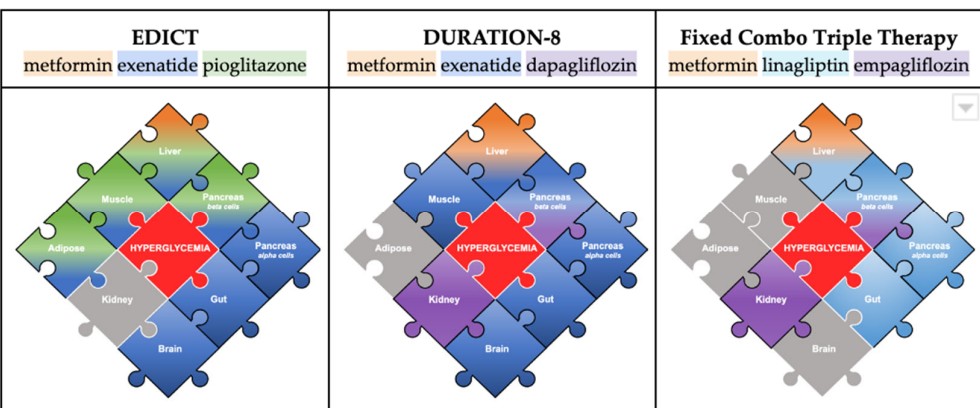

**Figure 3.** This figure highlights the pieces of the pathophysiology addressed by the trials EDICT and DURATION-8 compared to the use of a fixed-combination therapy (Trijardy). The colored gradients show the effect of specific drug classes on each of the pathways. The gray pieces indicate the pathways that are not addressed by that particular combination. In the center is the main treatment target: hyperglycemia.

### 6.1. Combination of Metformin, TZD, and GLP-1RA (7/8 of the Puzzle Addressed)

In 2015, the 3-year study Efficacy and Durability of Initial Combination Therapy for Type 2 Diabetes (EDICT) was published, which compared the efficacy, durability, and safety of initial triple therapy (metformin/pioglitazone/exenatide) versus conventional therapy [7]. The combination of metformin, a TZD, and a GLP-1RA provided a synergistic effect on the liver, pancreas (alpha, beta cells), muscles, adipose tissue, gut, and brain. This trio effectively addressed 7/8 pieces of the puzzle, with renal glucose reabsorption left as the only factor left untreated. Additionally, early initiation played an important role in achieving a significantly greater reduction in the HbA1c levels compared to the conventional therapy (5.95% vs. 6.50%, respectively; $p < 0.001$), a 7.5-fold lower rate of hypoglycemia, and a mean weight loss of 1.2 kg [48]. Beyond effective glycemic control, the combination demonstrated superior durability, as more patients in the intervention arm achieved the HbA1c goal of <6.5% and maintained it over three years, while patients receiving conventional therapy had a gradual increase in HbA1c levels [48].

### 6.2. Combination of Metformin, GLP-1RA, and SGLT-2i (7/8 of the Puzzle Addressed)

Another triple-therapy-combination study was DURATION-8. This 28-week study, later extended to 104 weeks, assessed the efficacy of exenatide (GLP-1RA) and dapagliflozin (SGLT-2i) simultaneously added to metformin monotherapy in patients with poorly controlled type 2 diabetes [49]. At week 28, the combination demonstrated superiority to both placebo arms in reducing HbA1c levels by up to 2%, and this effect was maintained at week 52 and week 104 [49]. This combination of metformin, a GLP-1RA, and an SGLT-2i provided additive effects on the liver, pancreas (alpha, beta cells), muscles, gut, brain, and kidneys. It also tackled only 7/8 pieces of the puzzle, including renal glucoprotective effects; yet, without a TZD, lipotoxicity remained untreated.

*6.3. Fixed-Combination Therapy: Metformin, DPP-4i, SGLT-2i (5/8 of the Puzzle Addressed)*

When considering pharmacologic therapy for type 2 diabetes, an additional aspect to consider is the increasing medication burden for patients due to comorbidities such as dyslipidemia, hypertension, and cardiovascular disease [50]. A common challenge clinicians face is determining whether the patient's hyperglycemia is due to poor medication adherence or is occurring despite medication compliance, which implies a need for treatment intensification [51]. A retrospective cohort study identified an increased pill burden and dosing frequency as major factors responsible for a significantly poorer adherence to oral therapy [52].

Another potential option for treatment is a fixed-drug combination (FDC). The FDA has approved several dual-therapy FDCs. Examples of these include the first oral diabetes triple therapy (empagliflozin, linagliptin, and metformin hydrochloride extended-release Trijardy$^R$) in 2020 seen in Figure 3. This combination completes 5/8 puzzle pieces (kidneys, pancreas (alpha, beta cells), liver, and gut) but cannot complete the puzzle. While there are no head-to-head efficacy trials with this triple therapy, existing trials have shown that the combination of empagliflozin/linagliptin as an FDC has been efficacious when combined with metformin [53,54]. The long-term effects of these combinations are yet to be determined, so no conclusion can be drawn on their durability. FDCs should be used cautiously due to their drug interactions and contraindications [52]. Furthermore, another drawback is the limited potential for dose titrations, making it challenging to achieve the perfect balance between efficacy and adverse events compared to separate combination therapies [52].

## 7. Ideal Combination

Considering existing trials and medications, the question remains: what is the ideal combination therapy to complete the entire octet puzzle?

To recap, the ideal combination therapy should be effective at treating all eight pathways: the liver (decrease in hepatic glucose production), pancreas (increase in insulin secretion, decrease in glucagon secretion), muscle (increase in glucose uptake), adipose (decrease in lipotoxicity), gut (increased incretin effect), brain (neurotransmitter regulation, decrease in appetite), and kidneys (decrease in glucose reabsorption) [55]. This is depicted in Figure 2, in which each drug class is color-coded based on the organ-related puzzle piece that it targets.

Based on the studies above, the combination that completes the octet puzzle is a TZD, GLP-1RA/twincretin, and SGLT-2i (Figure 4). TZDs (pioglitazone is the most commonly used) are beta-cell-protective and potent insulin sensitizers that exert effects on muscle, liver, and fat cells [28]. GLP-1RAs directly increase insulin secretion through beta-cell-protective functions and suppress glucagon secretions by inhibiting alpha cells and increasing insulin sensitivity [56]. Indirectly, they also cause slowed gastric emptying and weight loss and induce favorable lipid profiles [57]. Together, they can ameliorate 7/8 of the components of the octet. The missing piece is an SGLT-2i, which decreases glucose toxicity and exerts renoprotective effects.

The synergistic effect of this exact combination is yet to be determined; however, an indirect comparison can be made using the results of EDICT and DURATION-8. EDICT assessed the durability of metformin/TZD/GLP-1RA and DURATION-8 assessed the efficacy of metformin/GLP-1RA/SGLT-2i. Both trials showed significant improvements in glycemic control when compared to conventional therapy, and in particular, EDICT determined long-term durability with the triple-combination therapy [7,48,49]. However, both trials only tackled 7/8 pieces of the puzzle, indicating potential improvement.

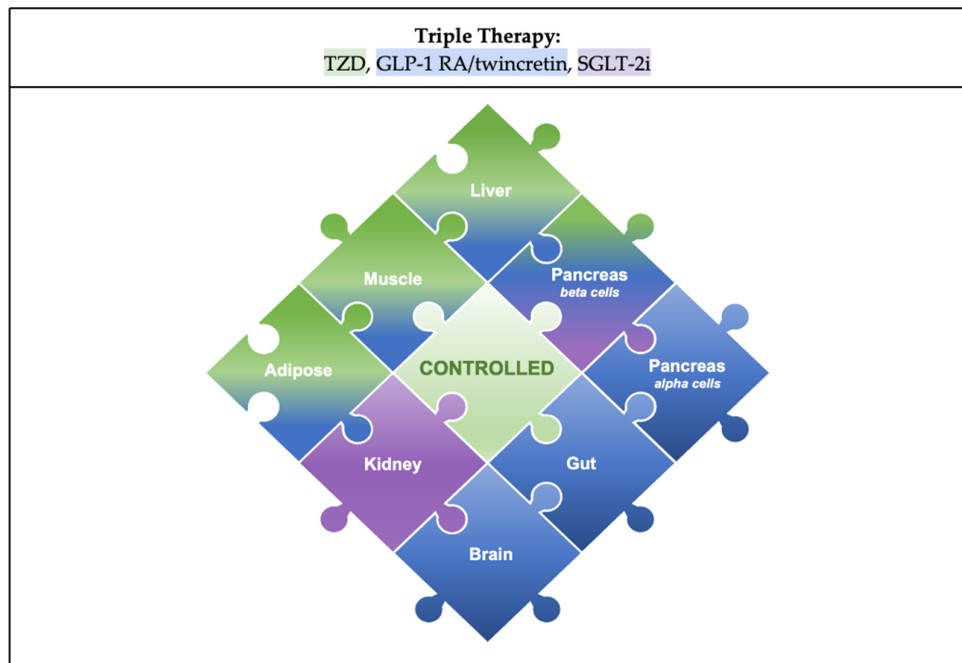

**Figure 4.** Proposed combination therapy of a TZD, GLP-1RA or Twincretin, and SGLT-2i. The colored gradients indicate which drug class addresses each piece: green for TZD, blue for GLP-1RA, purple for SGLT-2i. In the center is a new piece that signifies a controlled state of the disease, with the ideal combination addressing each of the eight pathophysiological mechanisms implicated in T2DM.

The medication replaced by both combinations is metformin. At the level of the liver, both metformin and TZDs are potent insulin sensitizers and effective at reducing hepatic glucose production [14]. However, at the level of muscle and fat, TZDs have more potent effects due to different physiologic mechanisms of action. In his trial EDICT, DeFronzo et al. reasoned that using both metformin and TZD creates an additive effect in muscle to further reduce the HbA1c [7,48,49]. However, the authors of this paper believe that choosing a medication that works to complete the entire octet rather than solely strengthen one puzzle piece will have a more substantial effect on glycemic control. For this reason, an SGLT-2i was chosen as the substitute medication.

## 8. Therapy Approach: Glucocentric or Cardio–Renal-Centric?

The octet theory highlights eight different pathophysiologic pathways that contribute to the disease progression of diabetes mellitus with hyperglycemia at the core [14]. In the early 20th century, the treatment protocol emphasized a glucocentric approach that primarily focused on maintaining glycemic control and reducing hyperglycemia [58]. However, epidemiologic trends have shown that rising cardiovascular–renal complications due to diabetes mellitus are the predominant cause of mortality [58]. Keeping this in mind, the treatment focus in the late 20th century shifted towards a cardio–renal-centric approach with organ-protective endpoints [59]. Physicians must choose medications that have more expansive effects than simply lowering hyperglycemia.

Among the antihyperglycemic agents, the two classes of medications that have been shown to mediate cardiovascular–renal protection are GLP-1RAs and SGLT-2is [59]. SGLT-2is have several extraglycemic effects, including 2–3 kg weight loss, persistent reductions in systolic and diastolic BPs, reduced lipotoxicity, the attenuation of factors associated with nonalcoholic steatohepatitis (NASH) and nonalcoholic fatty liver disease (NAFLD), natriuresis, which causes improved endothelial function leading to a decreased demand on cardiac tissue, and a reduction in intraglomerular pressure [29]. Similarly, GLP-1RAs have also been shown to have numerous secondary effects, including an associated weight loss of 3 kgs [60], reductions in systolic and diastolic BPs [61], improvement in lipid profiles [62],

the activation of anti-inflammatory pathways [63], and beneficial effects on NASH and NAFLD [64]. A systematic review of real-world studies determined that the initiation of SGLT-2is and GLP-1RAs yielded similar beneficial effects on composite CV outcomes, major adverse cardiac outcomes (MACEs), stroke, and MI, whereas SGLT-2is were more effective in the prevention of HF and all-cause mortality [65]. Additionally, there is an increasing amount of evidence on the potent synergistic effects of these medications when used in combination [66].

Aside from SGLT-2is and GLP-1RAs, the third component of the ideal combination therapy highlighted are TZDs. Although this medication class has no renoprotective effects, studies have shown that pioglitazone can lower BP and improve lipid profiles [67]. Furthermore, studies have determined pioglitazone to be an excellent agent for treating NASH/NAFLD, as it can reduce the hepatic fat content and reverse hepatic fibrosis [68]. However, pioglitazone has potentially harmful adverse effects, including increased fracture risk, edema, and heart failure [29,66]. The underlying mechanism of edema and HF is attributed to sodium reabsorption from the kidneys, and therefore, theoretically, this could be offset with SGLT-2is [29], but further research is required to substantiate this. Combination studies using SGLT-2is, pioglitazone, and metformin have shown reduced HbA1c levels and lowered BP, but weight gain and edema remain persistent [69]. As GLP-1RAs have been proven to have substantial weight loss effects [69], the use of a long-acting GLP-1RA in combination with pioglitazone could theoretically offset the weight gain. In the triple-combination therapy trial EDICT, the therapy arm did, in fact, have a mean weight loss of 1.2 kg; however, cardiovascular outcomes were not assessed in this trial, so there is no direct conclusion regarding the cardio–renal protection [7,48,49]. Therefore, in theory, a combination of GLP-1RAs, SGLT-2is, and TZDs could be a dual glucocentric and cardio–renal-centric approach with potential long-term macrovascular and microvascular benefits.

## 9. Limitations

While there are numerous potential benefits of utilizing this ideal treatment combination, this approach also has limitations. First, this combination has not been used in a formal clinical trial to assess its efficacy and adverse outcomes. However, the benefits of each medication and potential side effects are well described, and similar combination therapy studies as described above have demonstrated promising efficacy with reduced adverse outcomes compared to conventional therapy. In theory, the extraglycemic benefits of each medication can work to alleviate the side effects of the other medications when used in combination (for example, the weight loss benefit of an incretin-based therapy countering the weight gain from a TZD). Further investigation is warranted to assess the benefits and adverse effects of the combination suggested in this paper.

Second, the medication cost presents a practical limitation to widespread implementation. A recent study published in the 2022 Annals of Internal Medicine highlights the cost-effectiveness of using SGLT-2is and GLP-1RAs as first-line management for diabetes. Currently, it can cost approximately USD 20 per day to use SGLT-2is, and approximately USD 30 per day for GLP-1RAs. Per the study, SGLT-2is and GLP-1RAs are not cost-effective at their current rates, and they must be reduced to under USD 5 per day for SGLT-2is and under USD 6 per day for oral GLP-1RAs in order to be affordable [70]. Although this raises a fair concern in determining the affordability and accessibility of initial combination therapy, the downstream economic burden of diabetes complications must also be taken into account. Early, adequate glycemic control may decrease the lifetime cost of the disease, a majority of which is attributable to late-stage microvascular and macrovascular complications. While the initial cost of this combination therapy will be higher than conventional therapy, the possibility of significantly improved glycemic control and reduced complications may make it more affordable in the long term.

Finally, the limitations of medication adherence and polypharmacy must be considered. It is important to note that this combination may not be suitable for patients who

are unable or unwilling to take multiple medications. For patients who are hesitant to start multiple medications, thoughtful discussion about the benefits of adequate glycemic control may provide improved adherence. Additionally, many patients with diabetes will eventually require multiple medications after significant disease progression. By utilizing a combination therapy that addresses each of the eight pathophysiological mechanisms, it may be possible to slow the disease progression and subsequent complications. Additionally, the authors propose that early treatment with combination may allow for a reduction in long-term polypharmacy.

### 10. Back to the Case

In this particular scenario, our patient has multiple risk factors for the quick progression of the disease, such as dyslipidemia, obesity, an elevated waist ratio, and a low activity profile. Additionally, this patient has the added benefit of ample insurance and low copayments, which is a key limitation that is often faced with patients at the clinic. Starting the patient on the ideal combination of TZDs, GLP-1RAs, and SGLT-2is should be considered a viable option. Due to the new onset of the disease, with a combined medication effect, there is a chance to halt the disease progression before beta-cell exhaustion. Tackling all the pathways early on can help avoid the need for lifelong medication and potentially help patients gain true control over their glucose levels.

### 11. Conclusions

In conclusion, there is a wide spectrum of antihyperglycemic agents that can be utilized in the treatment of diabetes. Each of these agents addresses different pieces of the pathophysiological puzzle, but no monotherapy solves the puzzle alone. With studies demonstrating that traditional methods are insufficient, clinicians must reimagine the way in which diabetes treatment is approached. The eight-piece puzzle introduced in this manuscript can serve as a guide to help physicians decide which medication combinations will be most effective for their patients. Furthermore, the puzzle provides a framework for the creation of a comprehensive combination therapy that addresses all eight pathophysiological mechanisms at once. The combination of a TZD, GLP-1RA or twincretin, and SGLT-2i meets this criterion and has the potential to improve glycemic control, preserve beta-cell function, and provide organ-protective benefits. While further exploration into the efficacy of this combination is required, existing research on the pathophysiology of diabetes establishes this as a treatment that must be considered.

**Author Contributions:** R.G. contributed to the conception, design, drafting, and critical revision. A.S. contributed to the conception, design, drafting, and critical revision. J.H.S. contributed to the conception, design, drafting, and critical revision. All authors have read and agreed to the published version of the manuscript.

**Funding:** This research received no external funding.

**Institutional Review Board Statement:** Not applicable.

**Informed Consent Statement:** Not applicable.

**Data Availability Statement:** The data presented in this study are available upon request from the corresponding author.

**Conflicts of Interest:** R.G. has no relevant conflict to disclose. A.S. has no relevant conflict to disclose. J.H.S. has served as an advisor to Abbott, Astra Zeneca, Bayer, Eli Lilly, Nevro, and NovoNordisk.

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
