# Peer review of "Perfecting the Puzzle of Pathophysiology: Exploring Combination Therapy in the Treatment of Type 2 Diabetes"

_diabetology, doi:10.3390/diabetology4030032_

Round 1

Reviewer 1 Report

Please read the attachment. Thank you.

minor changes are needed.

Reviewer 2 Report

The  present submission tried to explore the best possible combination that  can have a durable effect on diabetes management on a pathophysiologic perspective. The idea is interesting. However, a few concerns rising about the current version.  

1) what is the meaning of Trigs in the abstract?

2) In Figure 1, why the liver stay close to Pancreas? In other word, can liver develop a direct link with muscle or gut? Please explain the orders of tissue in the puzzle.

3) In Line 171, what is the meaning of 49 in 'UK Prospective Diabetes Study (UKPDS) 49'?

4) In Lines 348-351, the authors stated that ' the extra-glycemic benefits of each medication can work to cancel out the respective side effects of the other medication when used as a combination' Please discuss the theory. My feeling is that the possibility is very limited. 

Round 2

Reviewer 1 Report

Dear Editor and Authors:

Thank you for providing the point to point response.

The manuscript has been improved significantly. It is could be accepted for publication.

Please feel free to contact me if you have further requests or concerns.

Thank you for reading